# Tenderness of PGI “Ternera de Navarra” Beef Samples Determined by FTIR-MIR Spectroscopy

**DOI:** 10.3390/foods11213426

**Published:** 2022-10-29

**Authors:** María José Beriain, María Lozano, Jesús Echeverría, María Teresa Murillo-Arbizu, Kizkitza Insausti, Miguel Beruete

**Affiliations:** 1Institute for Innovation & Sustainable Food Chain Development (IS-FOOD), Arrosadia Campus, Public University of Navarre (UPNA), Jerónimo de Ayanz Building, 31006 Pamplona, Spain; 2Institute for Advanced Materials and Mathematics (INAMAT2), Arrosadia Campus, Public University of Navarre (UPNA), Jerónimo de Ayanz Building, 31006 Pamplona, Spain; 3Antennas Group-TERALAB, Multispectral Biosensing Group, Navarrabiomed, Complejo Hospitalario de Navarra (CHN), Universidad Pública de Navarra (UPNA), Universidad Pública de Navarra (UPNA), IdiSNA, 31008 Pamplona, Spain

**Keywords:** FTIR-MIR spectroscopy, beef, *longissimus dorsi*, certified samples, tenderness

## Abstract

Understanding meat quality attribute changes during ageing by using non-destructive techniques is an emergent pursuit in the agroindustry research field. Using beef certified samples from the protected geographical indication (PGI) “Ternera de Navarra”, the primary goal of this study was to use Fourier transform infrared spectroscopy on the middle infrared region (FTIR-MIR) as a tool for the examination of meat tenderness evolution throughout ageing. Samples of the *longissimus dorsi* muscle of twenty young bulls were aged for 4, 6, 11, or 18 days at 4 °C. Animal carcass classification and sample proximate analysis were performed to check sample homogeneity. Raw aged steaks were analyzed by FTIR-MIR spectroscopy (4000–400 cm^−1^) to record the vibrational spectrum. Texture profile analysis was performed using a multiple compression test (compression rates of 20%, 80%, and 100%). Compression values were found to decrease notably between the fourth and sixth day of ageing for the three compression rates studied. This tendency continued until the 18th day for C20. For C80 and C100, there was not a clear change in the 11th and 18th days of the study. Regarding FTIR-MIR as a prediction method, it achieved an R^2^ lower than 40%. Using principal component analysis (PCA) of the results, the whole spectrum fingerprint was used in the discrimination of the starting and final ageing days with correct maturing time classifications. Combining the PCA treatment together with the discriminant analysis of spectral data allowed us to differentiate the samples between the initial and the final ageing points, but it did not single out the intermediate points.

## 1. Introduction

High levels of quality and safety in food production require strict quality assurance and process control. In turn, satisfying this demand requires appropriate analytical tools for food analysis both during and after production. Desirable features of such tools include speed, ease-of-use, minimal or no sample preparation, the avoidance of sample destruction, and low cost [1]. 

In the meat industry, it is important to control the sensory attributes of meat, namely appearance, juiciness, flavor, and texture. Surveys have shown that consumers have difficulty in selecting meat because they are unsure of its quality, particularly its texture. The texture is a multifaceted trait that encompasses qualities, such as cohesion, chewiness, springiness, and tenderness. [2]. Among them, tenderness has been identified as the key factor regarding consumers’ eating satisfaction for meat products. The condition of the myofibrils, the amount and type of collagen, and, in mature animals, the presence of intermolecular cross-links, all have an impact on how tender the meat is. The ageing procedure allows muscle endogenous enzymes to partially break down muscle tissue in the meat so it becomes more tender. The ageing conditions and time are key factors determining the texture characteristics of meat [3].

To control meat texture, instrumental techniques are usually employed. Tenderness is defined as the force required to attain a given deformation or penetration of a product [4]. It can be evaluated with testing panels or instrumental analysis, such as Warner–Bratzler [5], or by using a modified compression device [6,7]. These methods, however, have significant disadvantages, including complexity, time consumption, and sample destruction [8]. To overcome these testing constraints, the development of alternative techniques to assess the tenderness of meat and meat products is required. It is claimed that there may be a tight relationship between meat texture and structure. This is based on the assumption that internal meat structure and microstructure provide more information and further interpretation for textural modifications caused by external factors, such as ageing time. For the food business, Fourier transform mid-infrared (FTIR-MIR) spectroscopic techniques with data analytics stand out among the alternative techniques because they enable fast analysis times without destroying sample integrity [9].

Not as frequently used in food analysis as near-infrared spectroscopy (NIRS) is Fourier-transform mid-infrared (FTIR-MIR) spectroscopy. However, over the past two decades, there have been a lot more studies on the use of FTIR-MIR and chemometric instruments for food analysis. The structural characterization of food molecules, determination of the quality of raw materials and additives, and identifying food adulteration or authenticity are a few examples of general application areas [10]. Other general application areas include studying the interactions of food components. Using data fusion of attenuated total reflection (ATR) Fourier transform infrared (FTIR) spectroscopy and chemical characteristics, Nunes et al. [11] identified frauds in bovine meat to which non-meat components were added.

FTIR-MIR has also been applied on different types of food, such as grapes [12], olive oil [13], honey [14], tomato [15], potato [16], milk and dairy products [17], fish [18], and meat [19,20,21]. Regarding FTIR-MIR technology application to beef, Sinelli et al. [22] monitored fresh minced beef quality decay during marketing and Lozano et al. [23] showed the applicability of this technique to estimate the fat content of meat samples.

The PGI “Ternera de Navarra” covered product is fresh beef from calves born, raised, and slaughtered in Navarra, which meet all the requirements of the specifications. The PGI is fundamentally based on the Pyrenean bovine breed, native to the area, which currently provides around 90% of the meat covered. The bovine cattle of the Pardo Alpina, Blonde de Aquitaine, and Charolais breeds all adapted to the environment, and their crosses are also admitted. The feeding of the cattle will be adapted to the traditional norms of utilization of pastures in Navarra, according to the typical peculiarities that have marked meat production in these regions for centuries and that are linked to geographic and sociological factors typical of this community of Navarre. Breastfeeding will be mandatory, at least up to four months, and in the supplementary feeding, natural products and foods concentrates authorized by the Regulatory Council [24] will be used.

This research aimed to determine whether FTIR-MIR technology could be used to classify the tenderness of beef produced under the protected geographical indication (PGI) “Ternera de Navarra” according to differences based on the sample multiple compression test obtained at different ageing times. In addition, it was determined that wavelength range contributes to this meat tenderness differentiation.

## 2. Materials and Methods

### 2.1. Experimental Design

Figure 1 shows a schematic of the analyses carried out and the experimental strategy that was used to age beef meat.

### 2.2. Animal Material and Sample Maturing

Twenty young bulls from the “Ternera de Navarra” PGI in total were employed [24]. The animals were grown using conventional husbandry techniques, and after weaning at about six months of age, they were finished with concentrate feed and straw, both of which were given to them at no additional cost. The animals were transported from the farms to the official, commercial slaughterhouse on the day of the slaughter in accordance with international regulations governing the protection of animals used for scientific purposes [25]. In compliance with Council Regulation, EC, No. 1099/2009 [26] governing the protection of animals at slaughter, they were butchered at an abattoir in Pamplona (Spain). Just after bleeding, they were made into carcasses and hung from their hocks. The average weight and age at slaughter were 326 ± 24 kg and 379 ± 39 days, respectively. In accordance with EU Regulation 1249/2008, carcasses were rated for shape and fatness [27]. In addition, fatness values from 1 to 5 were transformed to an 18-point scale for SEUROP conformation, scoring 1 for P to 18 for S+, and a 15-point scale for fatness scores from 1 to 5, scoring 1 for 1- to 15 for 5+. The *Longisimus dorsi lumborum* (LDL) muscle was taken out of each carcass from the sixth to the eighth lumbar vertebrae after they were chilled for 72 h. Upon reception at the Public University of Navarre from the abattoir, the loins were vacuum packaged in polyamide/polyethylene (PA/PE) pouches (Vaessen Schoemaker Ind., Barcelona, Spain, 120 μm thick with an (oxygen) O_2_ permeability of 1 cc/m^2^/24 h, a CO_2_ (carbon dioxide) permeability of 3 cc/m^2^/24 h, an N_2_ (nitrogen) permeability of 0.5 cc/m^2^/24 h measured at 5 °C and 75% relative humidity (RH), and a water vapour transmission rate of 3 g/m^2^/24 h at 38 °C and 100% RH). Vacuum packaging (99% vacuum) was performed using a LERICA model C412 machine (San Giovanni Lupatoto, Italy). Then, they were aged for 4, 6, 11, and 18 days in the dark in climate-controlled chambers with circulated air at 4 °C and 85% relative humidity. Following aging, the samples were cut into slices and promptly frozen at 18 °C for later analysis. Prior to analysis, the samples were defrosted for 24 h at 4 °C.

### 2.3. Sample Analysis

#### 2.3.1. Proximate Composition

Steaks were thawed, their connective and surrounding adipose tissues removed, and then they were chopped into small pieces for grinding to determine the proximate composition (grinder Moulinex 800 W Dpa 251, Groupe Seb Iberica Ltd., Barcelona, Spain). Using the standard reference techniques for total fat [28], protein [29], and moisture content [30], the chemical compositions of the meat samples were assessed. Briefly, the Soxhlet extraction fat analysis was carried out in duplicate using heat and solvents. Briefly, 6 g of ground meat were digested for an hour in boiling HCl 3 M on a heating plate (Combiplac, JP Selecta, Barcelona, Spain). After that, the samples were filtered through filter paper (Albet 242), and they were dried at 70 °C for at least 12 h. Then, using the Soxhlet method with ethyl ether as the solvent, fat was extracted. The gravimetric variations of the round bottom flask, which held the extraction solvent during the Soxhlet cycles, were used to assess the fat content, and they were associated with the initial weight of the meat.

For protein analysis, 0.5 g of grinded meat were hydrolyzed with 15 mL of concentrated sulphuric acid at 390 °C for 5 h. After cooling, a neutralization with NaOH 0.1 M was done, and the resulting ammonium gas condensed in a vapour distillation system and was simultaneously titrated with HCL 0.1 M. A 6.25 conversion factor was used to determine the protein concentration, divided by the sample weight, and multiplied by 100 to obtain the protein percentage.

Moisture analysis was done as follows. Each homogenized steak was divided into ten grams, which were then weighed and placed in crucibles. To assess the moisture content, each sample was oven dried at 110 °C for 48 h. The results were expressed as a percentage of fresh meat.

#### 2.3.2. Texture Properties

A customized compression apparatus that prevents transverse elongation and assesses the mechanical characteristics of the samples’ muscle fibers was used to examine the instrumental texture of raw beef [31]. (Appendix A). The stress was assessed at three compression rates of 20%, 80%, and 100%. Samples, 1 cm^2^ in cross-section, were cut with muscle fibers parallel to the longitudinal axis of the sample. Eight replicates were performed per meat sample tested. Once obtained, the small cubes were kept at 4 °C for one hour. A Stable Micro System Model TA-XT2i texture analyzer (Stable Micro Systems, Godalming, England) was used in the tests. The test speed was 0.8 mm/s, and the strength employed was 0.045 N. A multiple compression cycle test was performed up to 20% (C20), 80% (C80), and 100% (C100) compression of the original portion height with a cylinder probe of 2 cm diameter. Force–surface deformation curves were obtained with a 25 kg load cell applied at a crosshead speed of 2 mm/s. Compression resistance or meat stress, which is defined as the force exerted per unit of contact surface of the sample when the different compression percentages have been applied, was measured in N cm^−2^. According to Sarriés and Beriain [32], the value at 20% compression is related to the strength of the myofibrils, while the value at 80% compression is related to breakage of the collagen tissue. This test also allowed assessing the stress resistance that was necessary to apply to break the structure of the meat, which corresponded to data obtained at 100% compression.

#### 2.3.3. Spectra Acquisition (Mid-Infrared Analysis)

FTIR-MIR spectroscopy was used to directly evaluate raw beef samples. For each beef sample analyzed, a total of six replicates were tested. A Bruker FTIR Vertex 80v spectrometer (Bruker Optik GmbH, Ettlingen, Germany) outfitted with a KBr beam splitter (400–10,000 cm^−1^), a Globar IR thermal source (operation bandwidth, 50–6000 cm^−1^), and a DLaTGS detector was used to gather the Fourier-transform infrared spectra (250–10,000 cm^−1^). All measurements were performed with an attenuated total reflectance accessory, A225/QPlatinum-ATR (Bruker, Ettlingen, Germany), with a diamond crystal. By combining 32 scans, all spectra were captured at a resolution of 4 cm^−1^ spanning the range v = 4000–400 cm^−1^. The ATR gadget was used to take a reference spectrum in vacuum. The spectrum of each sample was then determined. The samples were positioned on the ATR’s diamond crystal, making sure to completely cover the crystal’s surface. The ATR diamond crystal was cleaned with a piece of Berkshire Durx^®^ 670 optic cleaning wipe using distilled water and isopropyl alcohol after each sample measurement was completed so that it would be ready for the next sample’s reference measurement. This allowed it to be established that six repetitions of the same sample would be sufficient to produce reliable and comparable results. To avoid the spurious absorption peaks due to the atmospheric gases, measurements were done with the optical part of the instrument vacuumed. Moreover, to increase the precision and stability of the measurements performed, the instrument was placed in an ISO-7 cleanroom [33] with moisture (60%) and temperature controlled (21 ± 0.5 °C). 

### 2.4. Statistical/Chemometric Treatment

Statistical analysis was conducted using the IBM SPSS Statistics v.20 for Windows (SPSS Inc. Corporation, Hamlet, NY, USA) to obtain the descriptive statistics of the different compression rates tested in the samples. The means plus standard deviations of the stress values were obtained at the different ageing times. Statistical differences were assessed by performing an analysis of variance (ANOVA) to detect the influence of the ageing process. 

Moreover, a discriminant analysis was done to classify the samples by ageing using the spectral information. However, due to the high number of variables, before the application of this analysis, it was necessary to perform a principal component analysis (PCA) to reduce the number of variables. All these analyses were done using SPSS (v.20). 

The models were created using the chemometrics application, OPUS Quant 2. (Bruker, Ettlingen, Germany). Cross-validation was used to validate the calibration models developed using the partial least squares (PLS) regression technique. The key benefit of cross-validation was the reduction in sample requirements due to the use of the same set of samples for both method calibration and validation. One sample was removed from the group of samples prior to starting the calibration, and the system was calibrated using the remaining samples. The omitted sample was used to evaluate the model once it had been created. Once all samples had been utilized for validation, the cycle was restarted, this time with a different sample being separated. The optimum number of factors in the PLS calibration models was indicated by the lowest number that gave the minimum value of the root mean square error (RMSE) in cross validation to avoid overfitting the models. Using the OPUS Quant 2 application, the range of wavenumbers with the most information regarding their absorbance and the optimal pre-treatment technique were chosen for each variable under analysis.

#### Data Preprocessing

In each variable analyzed, the range of wavenumbers with more information in their absorbance and the best pre-treatment method were selected using the OPUS Quant 2 program. Data preprocessing can eliminate variations in offset or different linear baselines and is useful to ensure a good correlation between the spectral data and the compression values. In this study, different preprocessing methods were used. When working with the compression values, two preprocessing steps were applied. First, a whole calibration/validation model using the entire data set with all the studied samples was made. Since there were many samples, and to homogenize the spectral results, a second calibration/validation model with reduced data was also elaborated by taking the average of six repetitions for each sample and referred to as the mean set.

Two methods were used to get the compression results. The samples’ compression force curves were obtained using a straight-line subtraction method after the stress force meat was obtained using a derivative method that computes the first derivative of the spectrum. This method highlighted the peaks’ steep edges.

Finally, the idea of a more direct method of acquiring the textural properties of the samples was investigated to simplify the prediction models. The absorption values in the following wavelength ranges, which are associated with changes in the meat, were chosen as a result: 3200–2500 cm^−1^ and 2300–980 cm^−1^. The decision was made based on the literature (Appendix B) and data gleaned from an examination of the samples’ textural characteristics.

## 3. Results and Discussion

The animal carcasses were homogeneous, slightly fatty, and of high quality. The conformation of the carcasses was 95% U (very good) and 5% E (excellent) (European Rules) [27]. Indeed, when carcasses were classified by their fat cover, 100% of the samples were level 2 (slightly cover).

### 3.1. Proximate Composition of the Samples

The chemical composition of the samples was analyzed. The moisture and protein contents were 74.3% and 22%, respectively. The variation coefficients were 1.4% and 0.6%. The average intramuscular fat percentage of the samples was 2.5%, and the variation coefficient was 35.9%. The samples’ results for protein and moisture content were homogenous. However, the intramuscular fat content showed high variability that may be explained by the differences in the ages of the animals (374 ± 35). The moisture results agreed with those obtained by Beriain et al. [34] and Ripoll et al. [35] for beef from PGI-certified Ternera de Navarra with values ranging from 72.9% to 76.4% and 74.3% to 74.9%, respectively. Regarding the intramuscular fat content, the results in this study were higher than those obtained by Beriain et al. [34] (0.39–1.33%) with animal ages of 366 ± 23 days and those of Ripoll et al. [35] (1.14–1.57%), but still, they can be considered as low intramuscular fat content animals.

### 3.2. Tenderness by Multiple Compression Test

The connective tissue and myofibrillar network of meat muscle are both impacted by aging. Collagen fibers in connective tissue have been shown to play a role in preserving the muscle’s structural toughness and integrity. Table 1 shows the average and standard deviation (SD) values of the texture results obtained with the above-described compression method (C20, C80, and C100).

The C20 and C100 average compression values decreased during ageing, and significant differences were observed. The biggest changes in the tenderness occurred between the fourth and the sixth day of ageing. However, meat tenderness continued changing until the 11th day of ageing. The ageing process facilitates the action of endogenous proteases, mainly the calpain enzyme, which degrades the muscle proteins bonds and softens the meat [36]. For that reason, as they destroy the muscle structure, the strength necessary to compress the meat decreases [37]. Campo et al. [6] reported similar results, for four different beef breeds as regards the C20 value. They showed that beef meat stress at a compression rate of 20% was affected in a significant manner by ageing time, obtaining values in a range of 0.58–0.70 kg/cm^2^ for the initial day of ageing (day 4) and 0.42 to 0.51 kg/cm^2^ for the final day (18 days after slaughter). The results in this study were 0.41 and 0.31 kg/cm^2^ for day 4 and 18 of ageing, respectively, which were slightly lower than those reported by Campo et al. [6], mainly explained by the different ageing periods employed and the evaluation of different beef breeds. Ageing effect on the stress at a 20% compression rate and therefore on the myofibrils structure due to aging time showed significant differences in between the values obtained at 4 and 18 days.

The C80 mean values did not follow the same tendency. In this case, the strength necessary to compress the sample increased during the ageing when comparing the 4th day and the 18^th^ day. Novakofski and Brewer [38] described a similar effect named as the “paradox of toughening”. They detected that during the maturing process, the samples that were initially tender did not soften but usually became tougher. A reasonable explanation might be that ageing could alter the intrinsic water-holding capacity and the muscle structure, which could in the extreme case result in toughening, such as overcooking. However, Novakofski and Brewer [38] found that the “paradox” effect occurred in all the ageing sampling times which was not the case of this study. To understand these research findings, it is worth indicating that the intramuscular connective tissue is mainly composed of fibrillary collagen embedded in a matrix of proteoglycans (PGs), and their ratio influences meat tenderness achieved [39]. Similarly, enzymatic digestion during ageing can degrade PGs resulting in a higher exposure of collagen fibrils and can even lead to the formation of cross-linkages in collagen [40]. Collagen is very important in maintaining an acceptable texture; however, high amounts of crosslinks can greatly decrease tenderness and therefore give rise to C80 higher values compared to day four ageing time. Campo et al. (2000) [6] who performed an ageing study with beef from different breeds and evaluated tenderness with a similar compression system as was used in this paper study, concluded that compression values of raw samples at C80 compression were not affected by ageing, which, based on the above discussion of the results reported on this study, suggested that further elucidations of the muscle changed during chronological ageing of meat are needed.

Ripoll et al. [41] evaluated meat stress under the compression rates of 20% and 80% of 190 raw samples from 12 different beef breeds. They observed averaged values of 0.85 and 3.71 kg/cm^2^ for C20 and C80, respectively. The ranges for each parameter tested were 0.36–1.74kg/cm^2^ for C20 and 2.09– 7.34 kg/cm^2^ for C80, which were broad ranges due to microstructural differences in muscles compositions.

### 3.3. Assignment of the Representative Bands of the Beef FTIR Spectra

A representative spectrum of the samples analyzed are shown in Figure 2. In this spectrum, two wide peaks of high absorption intensity are highlighted (between 3700 and 3000 cm^−1^ and 980 and 400 cm^−1^), similar to the results obtained by Alamprese et al. [42]. These bands are associated with the vibration of the hydroxyl groups (O-H) which belong to the water. In these regions of the spectrum, other types of bonds also vibrate [43]. For example, around 3288 cm^−1^, the stretching vibration of the N-H bonds appears, typically from the protein amino acids. However, the high amount of water in the samples conceals other peaks of the spectrum, which become overlapped, and the differences between them are not noticeable. The symmetric and asymmetric stretching vibration of C-H bonds, which are typical of the methyl (CH3) and methylene (CH2) groups of the carbon chains of fatty acids, are associated with two peaks at 2925 and 2854 cm^−1^, respectively.

Appendix B contains the principal band assignment in the whole spectrum studied, and Figure 3 shows a detail of the spectrum in the region between 1900 and 900 cm^−1^. The region spectra profile is in correlation with those obtained by Sinelli et al. [22] for minced beef. Two peaks that appear around 1657 and 1542 cm^−1^ also stand out in the spectrum. The first is associated with the stretching vibration of alkene double bonds (C=C) and the N-H bond vibration of protein amino groups. The second, which combines vibrations of the N-H bond bending and the C-H bond stretching, is related to amides II. The remaining peaks in the spectrum are similarly significant even if their absorption intensity is less. The scissoring bending vibration mode of C-H bonds is linked to the peak at 1465 cm^−1^. The amino acid side chains and the carboxyl group (COO-) of fatty acids are related to the band at 1396 cm^−1^. The N-H bond bending vibration and the C-N bond stretching vibration are both related to the peak at 1239 cm^−1^. Similarly, the peak at 1162 cm^−1^ is connected to the stretching vibration of C-O bonds and the bending vibration of C-H bonds. The bending and twisting vibration of the fatty acids is responsible for the peak at 1117 cm^−1^.

The individual wavelengths that contributed the most to the data interpretation were grouped using PCA analysis to more easily interpret the spectrum results. For that, 1310 “wavelength” variables were used. The first two components’ matrix allowed for the identification of which ones were most important in explaining the overall variability. The first factor contributed 61.4% and the second 15.6%. A total of 231 wavelengths were excluded due to their low contributions to that variability.

### 3.4. Prediction Models

The partial least squares method (PLS) was used for constructing the tenderness prediction models, comparing the values obtained by the instrumental measurement with the spectral information. The OPUS Quant program was run to automatically select the data pre-treatment and the absorption intensities that give better results. Once the models were built, they were cross-validated using the leave-one-out method, taking out four spectra each time. Table 2 shows the results obtained for C20 and C80. It also includes the determination coefficient and the root mean square error for the prediction and validation model, the number of factors employed, the data-pretreatment used, and the wavelengths selected.

The highest percentage was obtained for the strength needed to compress the sample at 20% (38.54%) and the lowest (7.71%) was for the strength necessary to break the muscle structure (C80). These results are lower than 40%. Although the precision of the models is low, the result obtained for the C20 is like the ones achieved in previous studies made using near infrared (NIR) spectroscopy. For example, Ripoll et al. (2008) studied the chemical, instrumental, and sensorial qualities of the calf samples. To achieve this purpose, they analyzed samples belonging to different calf breeds and measured the tenderness using the Lepetit method and NIR spectroscopy. These authors obtained a determination coefficient of 32.9% for C20 and 38.6% for C80.

Prieto et al. [44] also evaluated the tenderness of calf meat, and they obtained a determination coefficient of 37%. For that reason, they concluded that the technique could be used as an early indicator, but it cannot replace the instrumental measurement of tenderness. Rahim et al. [8] used the NIR spectroscopy to build a prediction model to determine the chicken meat tenderness based on the type of feeding. These authors built a different model for each type of feeding analyzed, and they concluded that when chickens were fed ecologically, the tenderness could be predicted with high precision. The determination coefficient obtained varied between 85% and 95%, depending on the data pre-treatment employed. However, if the meat analyzed was from broiler chicken, the determination coefficients were lower, varying between 42% and 68%. Quiao et al. [45] assessed the use of VIS-NIR infrared spectroscopy for the prediction of beef quality parameters, including tenderness, using the slide shear force as the reference method. To achieve this purpose, these authors analyzed samples from the *M. longissimus thoracis* of 75 heifers, 118 steers, and 41 young bulls. The determination coefficients obtained during the validation process for the tenderness were 27.3%, 5.7%, and 15.0%, respectively. The authors concluded that although the predicted R^2^ values for tenderness were low, they were similar to the ones previously published. Sinelli et al. [22] showed that PCA applied to the FTIR-MIR spectra in the range of 1800–930 cm^−1^ had the ability to separate beef samples on the basis of “freshness” during storage under different conditions.

### 3.5. Classification of the Sample by Ageing

In the last part of the study, the classification of the samples by the ageing period was performed using discriminant analysis. The method employed was the step-by-step inclusion, and the validation was performed by cross validation. The absorption intensities between 1951 and 980 cm^−1^ were selected. However, due to the high number of variables used, it was necessary to first carry out a principal component analysis (PCA) to reduce them. The number of principal components employed was seven, which explained 99.108% of the initial variability. The wavelength range that contained the most information to allow the classification of the samples by the ageing appeared between 1400 and 1350 cm^−1^ where the stretching vibration of the COO- groups of fatty acids and amino side chains takes place [18,46].

Table 3 shows the results obtained in the classification of the beef samples in the different ageing days studied using the spectral information in the range between 1951 and 980 cm^−1^. The higher results of prediction percentages were obtained in the initial (4 days) and final (18 days) moments of ageing, with values of 44.2 and 53.3%, respectively. However, the results obtained in the intermediate periods of ageing were lower at 24.2% and 15.0% for the 6th and 11th days, respectively. Moreover, it can be observed that the samples with an intermediate period of ageing were not very different either between them or between the samples in the initial and final steps of the process. For that reason, nearly half (45.8%) of the samples aged for 11 days were classified in the group of 18 days of maturing. In the same way, 33% of the samples matured for six days were classified in the group of samples with a maturing period of four days. These results indicate that there is a change during the ageing process between the first two weeks of ageing and that it can be observed using FTIR-MIR spectroscopy. However, the period necessary to detect this change is large due to the different ways in which each meat aged, and two weeks is the time necessary for all the samples to reach the same tenderness level (Figure 4).

The same analysis was performed using only the samples in the initial and final steps of the process (4 and 18 days of ageing) (Table 4). Additionally, the absorption intensities between 1951 and 980 cm^−1^ were employed as variables of the discriminant analysis, and PCA was used to reduce the variables to seven principal components which explained 99.189% of the variability. For both ageing times, the percentage of correct classification was approximately 70%. These results are better than the ones obtained in the previous analysis. For that reason, it can be concluded that there exists a change in the molecular structure of the meat samples during the ageing process. This change can be observed by FTIR-MIR spectroscopy and enables the classification of the samples by using their spectral information.

## 4. Conclusions

The results obtained in this investigation show tenderness of meat belonging to the PGI “Ternera de Navarra”. PGI experienced the main increase between the fourth and sixth day of ageing, although it continued developing until the eleventh day. For that reason, to achieve higher meat tenderness it would be recommended to increase the ageing period until 11 days. The absorption spectra of the beef meat samples showed little differences at different ageing times. The determination coefficients of the prediction models obtained are lower than 40%, similar to the ones obtained in previous investigations. The principal component analysis (PCA) together with the discriminant analysis is useful to decide if the samples reach the optimal characteristics of tenderness.

## Figures and Tables

**Figure 1 foods-11-03426-f001:**
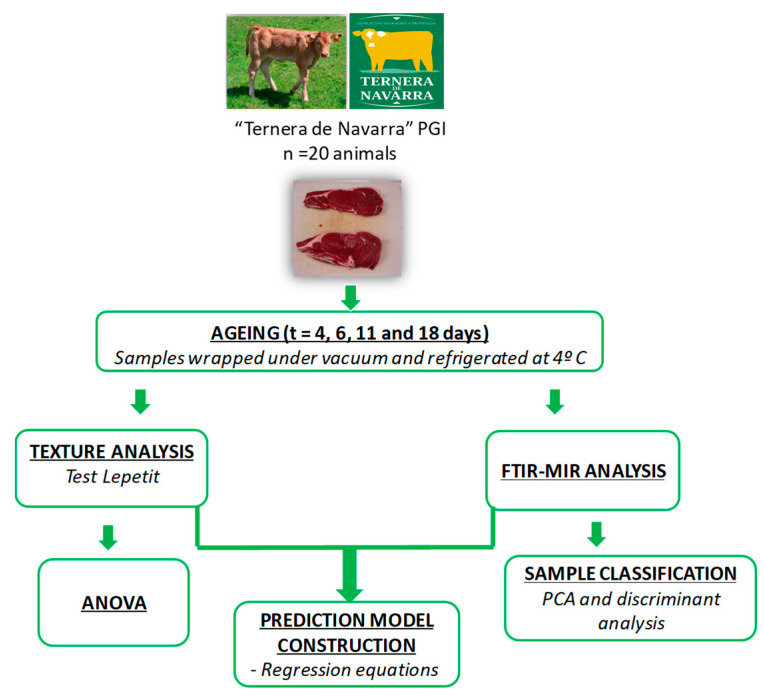
Scheme of the experimental design.

**Figure 2 foods-11-03426-f002:**
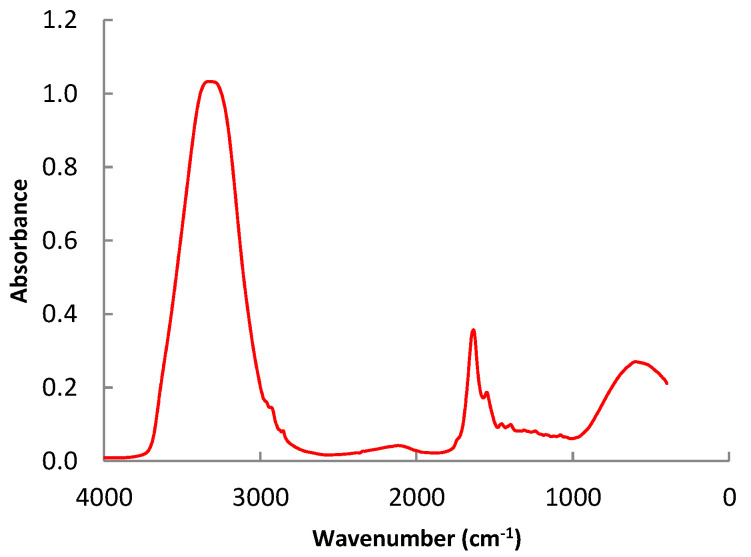
Representative spectrum of the samples analyzed.

**Figure 3 foods-11-03426-f003:**
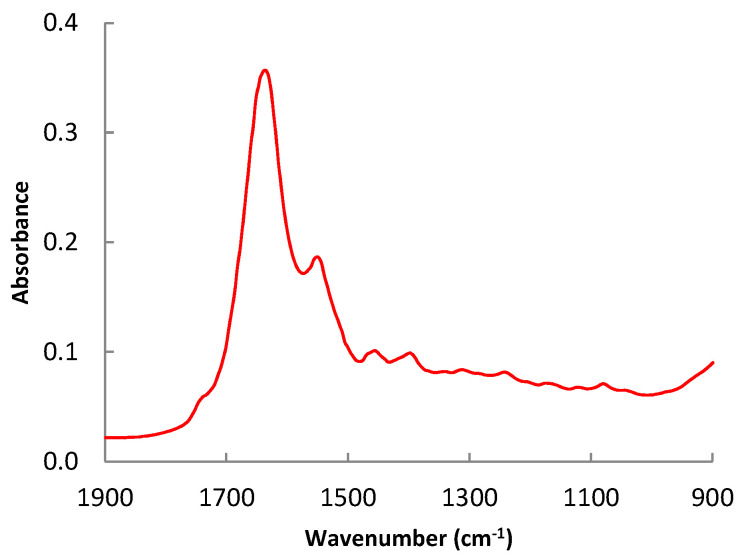
Detail of the representative spectrum in the region between 1900 and 900 cm^−1^.

**Figure 4 foods-11-03426-f004:**
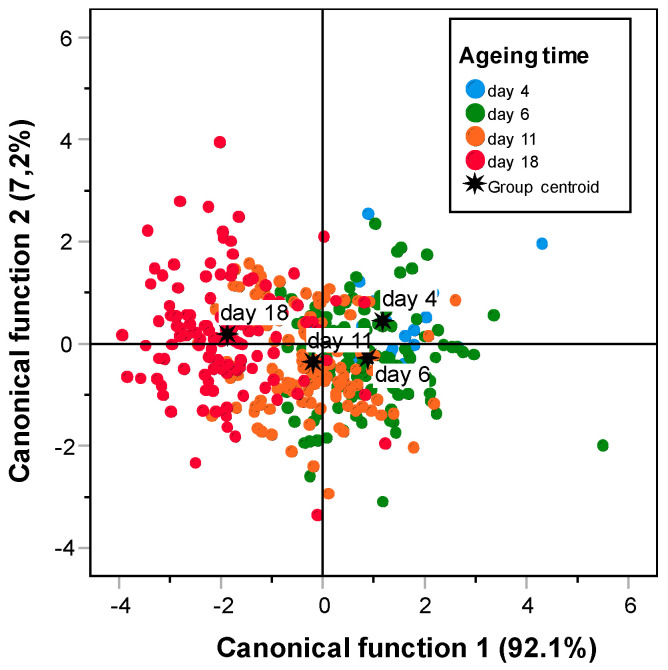
Classification of the samples according to the stepwise discriminant analysis. Of the 1079 wavelengths, 9 are enough to classify the samples (range 1951–980 cm^−1^).

**Table 1 foods-11-03426-t001:** Influence of the ageing on the averaged tenderness values. Average and standard deviation of the strength necessary to compress the meat samples at 20% (C20), 80% (C80), and 100% (C100) during the ageing.

Compress Percentage	Ageing (Days)				*p*-Value
	4	6	11	18	
C20 (kg/cm^2^)	0.41 ± 0.16 a	0.37 ± 0.14 b	0.33 ± 0.10 c	0.31 ± 0.08 c	**
C80 (kg/cm^2^)	3.45 ± 1.04 ab	3.30 ± 1.10 ab	3.29 ± 1.00 b	3.61 ± 1.09 a	*
C100 (kg/cm^2^)	9.42 ± 2.56 a	8.55 ± 2.79 b	8.71 ± 3.45 ab	8.36 ± 2.66 b	**

* *p*-value < 0.001; ** *p*-value < 0.05. Different superscripts in the same row correspond to the differences detected by ANOVA test.

**Table 2 foods-11-03426-t002:** Results of the prediction and validation optimized models for each variable analyzed.

Variables	Prediction		Validation			
	R^2^ (%)	RMSE	R^2^ (%)	RMSE	Factors	Data Pre-Treatment (Wavelengths Selected)
C20	52.27	0.052	38.54	0.059	9	First derivate (1951–1540 cm^−1^)
C80	35.00	0.667	18.24	0.724	7	First derivate (3200–2500; 2300–980 cm^−1^)
C100	14.77	2.030	7.71	2.110	3	Straight line subtraction (3200–2500; 1950–1540 cm^−1^)

**Table 3 foods-11-03426-t003:** Results obtained on the classification of the beef samples by the ageing (4, 6, 11, and 18 days) using the spectral information in the range between 1951 and 980 cm^−1^.

		Predicted Group				
	Ageing (Days)	4	6	11	18	Number of Spectra
**Original results (%)**	4	45.8	22.5	15.0	16.7	120
	6	32.5	25.0	14.2	28.3	120
	11	18.3	20.0	17.5	44.2	120
	18	15.8	14.2	15.8	54.2	120
**Cross validation results (%)**	4	44.2	24.2	15.0	16.7	120
	6	33.3	24.2	14.2	28.3	120
	11	19.2	20.0	15.0	45.8	120
	18	15.8	14.2	16.7	53.3	120

**Table 4 foods-11-03426-t004:** Results of the classification of the beef samples in the initial and final ageing days (4 and 18 days) using the spectral information in the range between 1951 and 980 cm^−1^.

		Predicted Group		
	Ageing (Days)	4	18	Number of Spectra
Original results (%)	4	71.7	28.3	120
	18	30.0	70.0	120
Cross validation results (%)	4	69.2	30.8	120
	18	30.0	70.0	120

## Data Availability

All data from the research conducted are available on request from the corresponding author.

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
