# Peer review of "Tenderness of PGI “Ternera de Navarra” Beef Samples Determined by FTIR-MIR Spectroscopy"

_foods, 2022, doi:10.3390/foods11213426_

Round 1

Reviewer 1 Report

 This study was conducted to evaluate beef tenderness using FTIR-MIR spectroscopy. The research objective is reasonable, and experimental approaches are well-fitted. However, some descriptions and discussion should be improved for further publication.

General comment

In this study, young bulls raised in a specific area, called Ternera de Navarra, were used. It has been known that the spectroscopic evaluation techniques for meat quality attributes is generally affected by the breed and age of livestock. In this regard, this reviewer wonders whether the results of this study using only samples from a specific region have a wide range of applications.

Minor comment

L83-89 It would be better to add more information on the PGI “Ternera de Navarra”.

L207 Add a period after [36], and check this point throughout the entire manuscript.

L283-304 In relation to the discussion, this reviewer suggests that the additional description of the cooking loss of the used samples may be useful to improve readers’ understanding.

L272 Delete parentheses indicating references throughout the entire manuscript.

Author Response

Comments and Suggestions for Authors

 This study was conducted to evaluate beef tenderness using FTIR-MIR spectroscopy. The research objective is reasonable, and experimental approaches are well-fitted. However, some descriptions and discussion should be improved for further publication.

Answer: Additional descriptions and discussion have been added following reviewer comments.

General comment

In this study, young bulls raised in a specific area, called Ternera de Navarra, were used. It has been known that the spectroscopic evaluation techniques for meat quality attributes is generally affected by the breed and age of livestock. In this regard, this reviewer wonders whether the results of this study using only samples from a specific region have a wide range of applications.

Answer: the authors understand the reviewer concern. However, the manuscript clearly reflects the scope of itself by indicating in the tittle and in the aim that the study is applied to the PGI “Ternera de Navarra, and, yes, it would be interesting to perform a similar study with samples from other regions.

Minor comment

L83-89 It would be better to add more information on the PGI “Ternera de Navarra”.

Answer: A new paragraph has been added in the introduction.

“The PGI “Ternera de Navarra” covered product is fresh beef from calves born, raised and slaughtered in Navarra, which meet all the requirements of the Specifications. The PGI is fundamentally based on the Pyrenean bovine breed, native to the area, which currently provides around 90% of the meat covered. Bovine cattle of the Pardo Alpina, Blonde de Aquitaine, and Charolais breeds, all of them adapted to the environment, and their crosses are also admitted. The feeding of the cattle will be adapted to the traditional norms of utilization of pastures in Navarra, according to the typical peculiarities that have marked meat production in these regions for centuries and that are linked to geographic and sociological factors typical of this Community of Navarre. Breastfeeding will be mandatory, at least up to four months, and in the supplementary feeding will be used natural products and foods concentrates authorized by the Regulatory Council [24].”

L207 Add a period after [36], and check this point throughout the entire manuscript.

Answer: we do not understand what the reviewer refers to. Please is it possible to provide with a better explanation?

L283-304 In relation to the discussion, this reviewer suggests that the additional description of the cooking loss of the used samples may be useful to improve readers’ understanding.

Answer, samples were not cooked for tenderness testing and therefore, description of the cooking lost is not applicable in the discussion.

L272 Delete parentheses indicating references throughout the entire manuscript.

Answer, the whole document has been checked in order to delete parentheses non applicable.

Authors additional comment: to avoid repetitions, several paragraphs of the manuscript have been re-written.

Authors additional comment: After figure 3 the following paragraph has been written.

“The individual wavelengths that contributed the most to the data interpretation were grouped using a PCA analysis to make easier the spectrum results interpretation. For that, 1310 "wavelength" variables were used. The first two components' matrix allowed to identify which ones were most important in explaining the overall variability. The first factor contributed with a 61.4% and the second with a 15.6%. A total of 231 wavelengths were excluded due to their low contribution to that variability.”

Authors additional comment:  thanks for the revision.

Reviewer 2 Report

The objective ot the presented work is very interesting and significant for the development of the discipline. The experiment is properly designed, however, considering a model study which plans a creation of prediction model/s should involve more samples than only 20... then the model will be more precise and acurate.

1. Why the sampleas of LD were frozen after maturation? Was not it better to analyzed them in fresh state, as freezing significantly influences meat structure and tenderness.

2. Why the connective and adipose tisseus were removed before proximate analysis. They are usually an integral part of the steaks presented to consumers?

3. What was the base of selecting day 4, 6, 11 and 18 of ageing to analytical plan? 

4. For the PCA it should be more reader friendly to present the results on graphs. 

Conclusions are clear and soundness.

Author Response

Comments and Suggestions for Authors

The objective of the presented work is very interesting and significant for the development of the discipline. The experiment is properly designed, however, considering a model study which plans a creation of prediction model/s should involve more samples than only 20... then the model will be more precise and accurate.

Answer: the authors do agree with the reviewer, however due to the lab capacity for samples managing, the number of 20 samples was the affordable quantity.

  1. Why the samples of LD were frozen after maturation? Was not it better to analysed them in fresh state, as freezing significantly influences meat structure and tenderness.

Answer: this is a very valuable comment that authors will take into account for future experiments. We support the conditions of the experiment described on the manuscript considering that the similar procedure was performed for all the samples and therefore if affected by the freeze step all of them were similarly affected.

  1. Why the connective and adipose tissues were removed before proximate analysis. They are usually an integral part of the steaks presented to consumers?

Answer: the connective and adipose tissue removed was that covering the steak on the surrounding which usually is not eaten by consumers. Indeed, the procedure followed was that recommended by ISO guidelines for proximate analysis of meat.

  1. What was the base of selecting day 4, 6, 11 and 18 of ageing to analytical plan?

Answer: except for breeds with hardness values higher than normal, the optimum ripening point occurs closed to days 18-21. In the experiment performed on this manuscript, we desired to obtain meat with different tenderness in order to study the capacity of MIR to differentiate them and therefore the chosen aging time were the indicated in the paper.

  1. For the PCA it should be more reader friendly to present the results on graphs.

Answer: following reviewer recommendation, figure 4 has been added to the manuscript

Conclusions are clear and soundness.

Authors additional comment: to avoid repetitions, several paragraphs of the manuscript have been re-written.

Authors additional comment: After figure 3 the following paragraph has been written.

“The individual wavelengths that contributed the most to the data interpretation were grouped using a PCA analysis to make easier the spectrum results interpretation. For that, 1310 "wavelength" variables were used. The first two components' matrix allowed to identify which ones were most important in explaining the overall variability. The first factor contributed with a 61.4% and the second with a 15.6%. A total of 231 wavelengths were excluded due to their low contribution to that variability.”

Author’s additional comment:  thanks for the revision.
